# Atmospheric Pollution in Port Cities

**Shnelle Owusu-Mfum [1], Malcolm D. Hudson [1], Patrick E. Osborne [1], Toby J. Roberts [2], Lina M. Zapata-Restrepo [2] and Ian D. Williams [2,***

[1] Faculty of Environmental and Life Sciences, University of Southampton, Highfield Campus, University Road, Southampton SO17 1BJ, UK; som1917@soton.ac.uk (S.O.-M.); mdh@soton.ac.uk (M.D.H.); peo@soton.ac.uk (P.E.O.)

[2] Faculty of Engineering and Physical Sciences, University of Southampton, Highfield Campus, University Road, Southampton SO17 1BJ, UK; t.j.roberts@soton.ac.uk (T.J.R.); lina.zapatar@udea.edu.co (L.M.Z.-R.)

* Correspondence: idw@soton.ac.uk

**Abstract:** Authoritative, trustworthy, continual, automatic hourly air quality monitoring is a relatively recent innovation. The task of reliably identifying long-term trends in air quality is therefore very challenging, as well as complex. Ports are major sources of atmospheric pollution, which is linked to marine traffic and increased road traffic congestion. This study investigated the long-term trends and drivers of atmospheric pollution in the port cities of Houston, London, and Southampton in 2000–2019. Authoritative air quality and meteorological data for seven sites at these three locations were meticulously selected alongside available traffic count data. Data were acquired for sites close to the port and sites that were near the city centre to determine whether the port emissions were influencing different parts of the city. Openair software was used for plots and statistical analyses. Pollutant concentrations at Houston, Southampton and Thurrock (London) slowly reduced over time and did not exceed national limits, in contrast to $NO_2$ and $PM_{10}$ concentrations at London Marylebone Road. Drivers of atmospheric pollution include meteorology, geographical and temporal variation, and traffic flow. Statistically significant relationships ($p < 0.001$) between atmospheric pollution concentration and meteorology across most sites were found, but this was not seen with traffic flows in London and Southampton. However, port emissions and the other drivers of atmospheric pollution act together to govern the air quality in the city.

**Keywords:** air pollution; port cities; time series plot; polar plot

## 1. Introduction

The majority of world trade is carried by sea, >80% at present, which is fundamental to the global economy [1]. The continued growth of international trade and globalisation of manufacturing processes has established ports as hubs for marine transport, vital to the development of many cities, especially port cities [2,3]. Port cities are urban areas that have a port present, which allows for the docking of ships, transfer of cargo and/or people to and back [4]. The presence of a port within port cities provides a different set of opportunities in comparison with non-port cities [5]. Major port cities experienced high levels of economic growth with a 13% increase in their aggregated Gross Domestic Product between 2008 and 2013 compared to only 8% with port cities in the same period. Ports provide both added value and employment through their direct activity, although with the latter, automation in port operation and cargo handling threatens this [6]. Renewable energy (i.e., tidal and wave), industrial development, tourism, linkages as well as culture and identity are some of the benefits associated with ports [5].

Ports are associated with adverse environmental impacts which includes destruction of marine habitat, introduction of non-native species and environmental pollution [5,7]. To be successful, future ports must operate more sustainably and in greater harmony with

their local population [8]. It is important that ports provide benefits for the populations of local cities to offset negative impacts [8]. Atmospheric pollution is a persistent challenge in especially urban areas that are hotspots for the emissions of atmospheric pollutants [9]. The major sources of atmospheric pollutants in urban areas are residential, road traffic, and industrial, as summarised in Table 1. However, marine traffic is a key source of atmospheric pollution in port cities compared to non-port cities, as ports are concentrated areas of marine transport [10]. Marine transport has been attributed to ~15% of global anthropogenic nitrogen dioxide (NOx) and 5–8% of sulphur oxide (SOx) emissions [1]. The presence of a port in these cities can worsen traffic congestion due to freight traffic to and from the port [11]. This is crucial as traffic congestion is associated with increased vehicle emissions, which are the dominant source of atmospheric pollutants [12]. Furthermore, ports can contribute to most of the total emissions of particular pollutants in a port city, as seen in Los Angeles with 45% of sulphur dioxide emissions attributed to the port. However, there have been significant efforts in many countries to curb emissions from other sources such as non-port-related road traffic, industrial and power generation compared to ports [13].

**Table 1.** Major sources of pollutants in port cities and atmospheric pollutants associated with these sources [14–19].

| Source | Pollutants |
| --- | --- |
| Road Traffic: From the combustion of diesel and petrol in engines, released into the atmosphere via vehicular exhaust. | Carbonaceous particulate matter (PM) including Black Carbon, Volatile Organic Compounds (VOCs), Primary Organic Aerosols. Inorganic compounds: trace Nitrogen Oxides ($NO_x$), Ammonia and Sulphur dioxide ($SO_2$) |
| Road Traffic (Non-exhaust): Including brake wear, tyre, road surface wear and resuspended road dust. | $PM_{10}$ and $PM_{2.5}$ |
| Marine Traffic including shipping and cruise ships: From combustion of low-grade fuel that is rich in sulphur, which occurs in ship engines. | Nitrogen oxide ($NO_x$) including NO and $NO_2$. Sulphur Oxides ($SO_x$) including $SO_2$ and $SO_3$. PM including organic carbon, black carbon, polycyclic hydrocarbons (PAH) and heavy metals |
| Residential: Combustion of fuel for heating and cooking. | $SO_x$, PM, $NO_x$, heavy metals, VOCs, and PAHs |
| Industrial | $SO_x$, PM, $NO_x$, heavy metals, and non-methane volatile organic compounds |

Atmospheric pollution is widely acknowledged as a major health hazard globally, as it attributes to around 7 million premature deaths annually [20,21]. Nitrogen oxides, sulphur dioxide, particulate matter, metals, and volatile organic compounds are the main pollutants seen in atmospheric pollution [22]. Long-term exposure to these pollutants has been associated with the increased risk of chronic diseases, including chronic obstructive pulmonary disease (COPD), cardiovascular disease and cancer ([22,23]; Manisaldis et al., 2020). Human health impacts associated with atmospheric pollution are projected to worsen in coastal cities as population is estimated to reach 10 billion by 2050, with coastal cities expected to see most of the growth [5,24]. The concentration of atmospheric pollutants to which individuals in a port city are exposed to is dependent on emissions, meteorological conditions, dilutions, and transformations [25]. Whether pollution concentrations are increasing or decreasing over the long term is a frequently asked question. Shorter term variations frequently conceal trends, or make analysing them difficult. Automatic monitoring of air pollution started in a few developed countries the early 1970s, but there are very few sites with a data record that extends that far. In most cases, a continual data record for key air pollutants only extends back to the mid–late 1990s. The task of reliably identifying long-term trends can therefore be very challenging. The emissions of atmospheric pollutants in port cities vary between cities of various sizes; this is influenced by the level of port activities and the urban population of the city itself [5]. Therefore, it is crucial to understand how these drivers influence atmospheric pollution in port cities, and the aspects of the

ports that are the most detrimental to atmospheric pollution in comparison with emissions associated with activities related to the city itself.

The aims of this study were to:

- Identify and review drivers of atmospheric pollution in the port cities of Houston, London, and Southampton; and
- Establish the long-term trends and evaluate the impact(s) of these drivers on atmospheric pollutant concentration in these port cities.

## 2. Materials and Methods

### 2.1. Study Areas

The City of Houston ($29°45'16°$ N and $95°22'59°$ W) is located in the Greater Houston metropolitan area, USA [26]. It is the fourth largest city by population in the USA, with a population of 2,325,502 and covering an area of 1722 $km^2$ [26]. The City of Houston is a major transport hub with three airports that has suitable rail links to Southern, Midwestern, and Western USA [27]. The Port of Houston is an important asset in the area and ranks as the largest port for international tonnage and second for overall tonnage [27]. The port is around 40 km in length and encompasses part of the Houston Ship Channel [28].

Three study sites were selected for data collection in and around the city's boundaries, which are Houston Aldine, Houston Baytown, and Lynchburg Ferry (Supplementary Materials, Figure S1). Houston Aldine is located near the north of the city. The monitoring station is located near the major Aldine Mail Route Road [29,30]. Aldine was selected as the non-port site due to its proximity to the city and availability of sufficient long-term authoritative data. Baytown and Lynchburg Ferry were selected as the port sites due to their proximity to the Houston Ship Channel [31]. $NO_2$ and $PM_{2.5}$ concentrations have been monitored in Aldine, whilst in Baytown, $PM_{2.5}$ was monitored, and $NO_2$ was monitored in Lynchburg Ferry, so they were both selected to enable suitable comparisons with Aldine.

Greater London, UK ($51°30'26''$ N and $0°7'39''$ W) is the capital and largest city of the UK, with a population of 8,961,989 and an area of 1572 $km^2$ [32]. London is a global city and its economy is dominated by service industries [33]. London is the dominant transport hub in the UK as it is the centre of road and rail networks; also, Heathrow International Airport makes it a crucial international transport hub [33]. The Port of London is one of the largest ports in the UK, with 54 million tonnes of freight in 2019 and 10 million journeys annually [34]. The Port of London covers 153 km of the River Thames stretching from Teddington to a defined boundary (from Foulness point in Essex via Warden Point in Kent) with the North Sea [35].

Two study sites were selected for data collection in and around the city's boundaries: London Marylebone Road and Thurrock (Supplementary Materials, Figure S2). London Marylebone Road is located in Central London, and experiences high air pollution due to congestion and high traffic flows [36,37]. The monitoring station is adjacent to the A501 London Marylebone Road [38]. The monitoring station in Thurrock is located near the A126 London Road, which is a main road in the area [39]. The site is 400 m north of the River Thames and is in proximity to the Port of Tilbury, which is the main port in the Port of London [39]. London Marylebone Road and Thurrock were selected as the non-port and port sites, respectively, as sufficient long-term authoritative data are available.

Southampton, UK ($50°54'$ N and $1°24'$ W) is the largest city in Hampshire that has a population of 256,459 and an area of 72.8 $km^2$ [40]. The city is served by an international airport and is well connected to London via the train network and motorway (M27) [41]. Manufacturing, retail, academia, and the Port of Southampton are vital industries in the area [41]. The Port of Southampton is the largest port in the UK by tonnage and is the main port for automotive trade and trade [42]. The port comprises four areas: Eastern Docks, Western Docks, Marchwood Industrial Park and Cracknore industrial Park, and a strategic land reserve for future port expansion known as Dibden Bay [43].

Two study sites selected for the study are within the city's boundaries, which are Southampton Centre and Southampton A33 (Supplementary Materials, Figure S3). South-

ampton Centre air quality monitoring station is located on Brinton's Road, and 20 m away from A3024 Bursledon Road which is a main road connecting the city centre to Port of Southampton's Eastern docks [44,45]. The Southampton A33 air quality monitoring station is located 5 m away from A33 Redbridge Road [44,46]. Southampton Centre was selected as the non-port site as it has sufficient long-term authoritative data available. Southampton A33 was selected as the port site, as it is in proximity to the Port of Southampton Western Docks [43].

### 2.2. Data Collection

Daily air quality data for 1 January 2000–31 December 2019 from Houston sites for $NO_2$ and $PM_{2.5}$ was acquired from the US Environmental Protection Agency (EPA) website as CSV files [47]. The daily maximum (max) $NO_2$ concentration collected for Lynchburg Ferry was only available from 1 January 2004. The hourly air quality data for 1 January 2000–31 December 2019 from London and Southampton sites for $NO_2$, $SO_2$, $PM_{10}$ and $PM_{2.5}$ was acquired from Defra's UK AIR website [48]. Southampton A33 only had air quality data from 1 January 2016 to 31 December 2009. The import AURN function in the Openair package for the statistical software R was used for this, where data for the sites were loaded into R as dataframes [49].

Meteorological data for the sites were acquired from World Weather Online between 1 July 2008 and 31 December 2019 [50]. It is forecasted data that are based on raw data from meteorological organisations, e.g., World Meteorological Organization [50]. The data included temperature, humidity, wind speed, and wind direction which were averaged over daily and hourly periods for Houston sites and UK sites, respectively. Daily pressure and temperature data for 1 January 2000–31 December 2019 from George International Continental Airport (37 km north of downtown Houston) were acquired from the National Centres for Environmental Information for Houston Sites [51,52].

Traffic counts data from the Department of Transport's Road traffics statistics website were collected for London and Southampton monitoring stations between 7:00 and 18:00 h on weekdays, at roads less than 5 min away by foot so that data were representative [53]. For London Marylebone, traffic counts were collected at the A501 London Marylebone Road in April, May, June, and September 2004–2019. Traffic counts for Thurrock were collected at A126 London Road in May 2001, March 2017, and April 2009. Traffic counts for Southampton Centre were collected at A3024 road in April, May, June, July, and October from 2001 to 2016. Traffic counts for Southampton A33 were collected at A33 Redbridge Road for September 2016, June 2017, and May 2019.

### 2.3. Data Processing and Analysis

The $NO_2$ concentration data for Houston sites were in parts per billion (ppb) which is a volume mixing ratio [54]. Therefore, the $NO_2$ concentration data were converted to micrograms per cubic metre of air ($\mu g\ m^{-3}$) using a standard equation [55]. The $NO_2$ data were converted to $\mu g\ m^{-3}$ to maintain consistency with the rest of the data used in this study. This involved using daily temperature and pressure data from the George Bush Intercontinental Airport, which were converted using standard equations [56].

The csv files containing pollutant concentrations for each year at Houston sites were merged into three dataframes containing the pollutant concentration between 1 January 2000 and 31 December 2019 using the plyr package in R. The datasets had missing dates, so they were inserted using the dplyr package. The dates were character string as they were downloaded as CSV files, so they were converted date objects using the asDate function in order to use the Openair package [49].

Openair was used to create time-series plots and time variation plots showing spatial and temporal trends in atmospheric pollutant concentrations across the study sites. Polar plots were created to characterise the relationship of atmospheric pollutant concentration with wind speed and direction. R was used to produce descriptive statistics for the atmospheric pollutant concentrations and meteorological data to analyse trends across

the years. The air quality and meteorological data for the sites were tested for normality using Q-Q plots and the Kolmogorov–Smirnov test for normality. The traffic counts were tested for normality using the Shapiro–Wilk test. The Spearman's Rank Correlation test in R was used to assess the relationship between atmospheric pollution concentrations and meteorological data and atmospheric pollution concentrations with traffic counts (London and Southampton only). The air quality data used in the correlation analysis with the traffic data were collected an hour after the traffic data to account for time lag for the pollutants to disperse from emission sources and reach the monitoring stations [57].

## 3. Results

### 3.1. Temporal and Spatial Patterns of Atmospheric Pollutant Concentrations

#### 3.1.1. Houston Sites

The trends in the atmospheric pollutant concentration in Houston's cannot easily be compared to those in the UK sites, as they were averaged over a 24 h period compared to a 1 h period for UK sites. The pollutant concentrations across the sites declined 2000–2019 (Figure A1). The highest average daily max $NO_2$ per year for Aldine was 69.9 $\mu g\ m^{-3}$ seen in 2000, and it was higher than that of Lynchburg Ferry with 54.9 $\mu g\ m^{-3}$ seen in 2018 (Supplementary Materials (A large amount of supplementary data has been made available. It is signposted in the text by the use of the prefix "S" in front of numbered tables and figures), Table S1). The lowest average daily max $NO_2$ per year for Lynchburg Ferry is 39.4 $\mu g\ m^{-3}$ seen in 2018, and it was higher than that of Aldine with 32.3 $\mu g\ m^{-3}$ in 2019 (Table S1). Additionally, average pollutant concentration per year tended to be higher for Aldine compared to Baytown and Lynchburg Ferry, except the average $NO_2$ concentration per year after 2009, which was higher in Lynchburg (Table S1). The $PM_{2.5}$ concentration exceeded the 24 h national limits in 2000–2002, 2008, 2012, 2014 and 2019 for Houston Aldine, which was also seen in 2000–2003, 2005 and 2019 for Baytown (Figure A1). The average $PM_{2.5}$ concentration for each year exceeded the 1-year national limits from 2000–2009 in Aldine and 2000–2005 in Houston Baytown (Table S1). The highest average $PM_{2.5}$ concentration per year was lower at Baytown with 12.9 $\mu g\ m^{-3}$ seen in 2000, 2003 and 2005, compared to Aldine with 13.8 $\mu g\ m^{-3}$ in 2000 (Table S1). The lowest average $PM_{2.5}$ concentration was seen in both sites in 2016, but it was lower at Baytown with 7.6 $\mu g\ m^{-3}$ compared to 8.4 $\mu g\ m^{-3}$ in Aldine (Table S1). However, the highest concentrations for $NO_2$ were seen in the winter months, whilst the lowest concentrations were observed in summer for Aldine and Lynchburg Ferry (Figure A2). The concentrations for $NO_2$ declined sharply across the weekend compared to the weekdays. The $PM_{2.5}$ concentration peaked in the summer months across the sites. The concentration of $PM_{2.5}$ in both sites was quite variable across the week and peaks on Saturdays.

#### 3.1.2. London Sites

The hourly mean $NO_2$ concentration for London Marylebone Road experienced some decline in 2000–2019; it did not deviate much in Thurrock (Figure A3). The $NO_2$ concentrations at London Marylebone Road exceeded the UK air quality objective (200 $\mu g\ m^{-3}$) for the 1 h mean $NO_2$ concentration except in 2002 and 2019 (Figure A3). The $NO_2$ concentration in Thurrock did not exceed the objective for $NO_2$, even though it rose above the 200 $\mu g\ m^{-3}$ in 2004 and 2008 as this only occurred once in both years (Figure A3). The hourly $NO_2$ concentration must exceed 200 $\mu g\ m^{-3}$ more than 18 times within each year to exceed the national objectives, which was not case in Thurrock (58). The average $NO_2$ concentration per year was highest in 2008 and 2003 for London Marylebone and Thurrock, respectively, but it was higher at London Marylebone with 115.3 $\mu g\ m^{-3}$ compared to Thurrock with 38.3 $\mu g\ m^{-3}$ (Table S2). The lowest average $NO_2$ concentration per year at both sites was seen in 2017, but London Marylebone Road was higher with 62.7 $\mu g\ m^{-3}$ compared to 23.4 $\mu g\ m^{-3}$ seen in 2003 (Table S2).

The hourly mean $PM_{10}$ concentrations do not deviate much across the sites (Figures A3 and A4). The $PM_{10}$ concentration peaked in 2002 at 400 $\mu g\ m^{-3}$ for London

Marylebone Road; the peak seen at Thurrock in 2003 was higher at 600 μg m$^{-3}$ (Figure A4). The highest average PM$_{10}$ concentration per year in both sites was seen in 2003, but it was higher for London Marylebone with 48.3 μg m$^{-3}$ compared to Thurrock with 29.9 μg m$^{-3}$ (Table S2). The lowest hourly average PM$_{10}$ per year was 22.5 μg m$^{-3}$ and 17.1 μg m$^{-3}$ seen in 2019 and 2015 for London Marylebone and Thurrock, respectively (Table S2). The daily mean PM$_{10}$ concentration rose above 50 μg m$^{-3}$ across both sites, but in Thurrock, it did not exceed the limits as it did not occur more than 35 times within each year (Figures A3 and A4) (Defra, 2021f).

The hourly mean SO$_2$ concentration at London Marylebone Road did not deviate much, but in Thurrock, it declined (Figure A4). The hourly SO$_2$ concentrations for London Marylebone and Thurrock was below 200 μg m$^{-3}$, except in 2004 for Thurrock where it peaked at 400 μg m$^{-3}$ (Figure A4). The SO$_2$ concentration in both sites was within the limits for SO$_2$ as it did not exceed 350 μg m$^{-3}$ more than 24 times within each year (Figure A4) (Defra, 2021f). The highest average SO$_2$ concentration per year was seen in 2000 for both sites; for London Marylebone, it was higher than for Thurrock, with 14.8 μg m$^{-3}$ and 8.6 μg m$^{-3}$, respectively (Table S2). The lowest average SO$_2$ concentrations per year were 3.1 μg m$^{-3}$ in 2018 and 0.9 μg m$^{-3}$ in 2019 for London Marylebone and Thurrock, respectively (Table S2). The hourly PM$_{2.5}$ concentration did not deviate much for London Marylebone Road, although it peaked above 800 μg m$^{-3}$ in 2001 (Figure A4). The highest and lowest average PM$_{2.5}$ concentrations per year were 16.7 μg m$^{-3}$ and 9.3 μg m$^{-3}$ seen in 2018 and 2001, respectively (Table S2).

The concentration of pollutants at the London sites experienced a sharp increase from around 05:00 to 09:00. At London Marylebone Road, the NO$_2$ continued to increase slowly after 09:00 until it declined at just before 18:00 (Figure A5). PM$_{10}$ experienced a slight decline after 12:00 compared to SO$_2$ and PM$_{2.5}$ which did not deviate much after 09:00 (Figure A5). The concentration of NO$_2$ and PM$_{10}$ at Thurrock dipped at 12:00; rose between 14:00 and 18:00 for NO$_2$ and 15:00 and 20:00 for PM$_{10}$ (Figure A5). The SO$_2$ concentration peaked just before 12:00, then it remained quite low and constant (Figure A5). For all the pollutants across the sites, the concentration was higher during the weekdays as opposed to the weekends and usually peaking on Thursdays and Fridays (Figure A5). The concentration of NO$_2$ for London Marylebone did not change across the seasons, but it was highest and lowest in November and August, respectively (Figure A5). The PM$_{2.5}$ and PM$_{10}$ concentrations were highest in February and March and lowest in June and July (Figure A5). SO$_2$ concentration remained below 20 μg m$^{-3}$ across the seasons, and in Thurrock, it also did not change much across the seasons (Figure A5). In Thurrock, the NO$_2$ concentrations were highest during the winter months and dipped between June and July. PM$_{10}$ in Thurrock dipped from July to August, but it was highest in early spring.

### 3.1.3. Southampton Sites

The hourly mean NO$_2$ concentration for Southampton Centre did not change much, but it declined in Southampton A33 (Figure A3). It was within the national objectives in both sites as opposed to those of London Marylebone (Figure A3). The peak average NO$_2$ concentration per year in Southampton Centre was seen in 2000 with 39.3 μg m$^{-3}$, and it was lower than Southampton A33 with 43.3 μg m$^{-3}$ in 2016 (Table S3). The lowest average NO$_2$ concentration per year for the sites was in 2019, but it was higher in Southampton A33 with 32.5 μg m$^{-3}$ compared to Southampton centre with 27.8 μg m$^{-3}$ (Table S3). The highest and lowest average NO$_2$ concentration per year was lower in the sites compared to London Marylebone but higher than that in Thurrock (Tables S2 and S3).

The hourly mean PM$_{10}$ concentration peaks at 400 and 300 μg m$^{-3}$ were observed in 2002 and 2016 for Southampton Centre and Southampton A33, respectively (Figure A4). The highest average PM$_{10}$ concentration per year was 27.7 μg m$^{-3}$ in 2003 for Southampton Centre, which was lower in Southampton A33 with 21.6 μg m$^{-3}$ in 2016 (Table S3). The lowest average PM$_{10}$ for Southampton Centre was 16.5 μg m$^{-3}$ in 2015, which was higher than that of Southampton A33 with 17.1 μg m$^{-3}$ in 2019. The peak and lowest average

$PM_{10}$ concentrations per year in both sites were lower than those of London Marylebone (Tables S2 and S3). The daily mean $PM_{10}$ concentration rose above 50 μg m$^{-3}$ in sites, but it was within the national limits since it did not occur >35 times in each year (Figure A3) [58].

The hourly mean $SO_2$ concentration in Southampton Centre experienced a decline, but the $PM_{2.5}$ did not change much (Figure A4). Similar to London sites, the $SO_2$ concentrations did not exceed the national objective, as the concentration did not rise above 350 μg m$^{-3}$ > 24 times within each year (Figure A4) [58]. The highest average $SO_2$ concentration per year was seen in 2000 with 8.3 μg m$^{-3}$, and the lowest was seen in 2017 at 1.4 μg m$^{-3}$ (Table S3). The highest average $PM_{2.5}$ concentration was seen in 2011 with 15.8 μg m$^{-3}$, and the lowest was seen in 2019 at 9.6 μg m$^{-3}$. The highest and lowest average per year for both pollutants were lower than those of London Marylebone.

The concentration of all pollutants measured at both sites declined over the weekend, which was also seen in Houston and London; $SO_2$ in Southampton Centre was an exception (Figures A3 and A5). $NO_2$ and $PM_{10}$ in both sites experienced an increase from 05:00 to 08:00 from Monday to Saturday, which was also seen in the London sites but not on Saturdays (Figure A5). The $NO_2$ concentration dipped at 12:00 but increased from 14:00 to 19:00 at both sites across the week (Figure A5). It was highest in both sites during the winter months and lowest in the summer months, similar to London (Figure A5). The spring months in both sites experienced the highest concentration of $PM_{10}$ and the lowest was seen in summer months, similar to London (Figure A5). The $PM_{2.5}$ at Southampton Centre followed this and it dipped at 11:00 and increased after 18:00 during the week, but in London Marylebone it did not deviate much after 7:00 (Figure A5). The $SO_2$ concentration at Southampton Centre did not deviate much across the seasons, between the days and the hours, similar to Thurrock (Figure A5).

*3.2. Traffic Counts and Atmospheric Pollutant Concentrations*

There were mainly weak positive relationships between hourly pollutant concentration and traffic counts across the sites. The strongest relationship was found between the $SO_2$ concentration and traffic counts at London Marylebone Road on 28 September 2009, and the lowest relationship was seen between the $PM_{10}$ concentration and traffic counts at Thurrock on 18 May 2001 (Table S4). However, most of the pollutant concentrations recorded at both London and Southampton sites did not have a significant relationship with the traffic counts at $p > 0.05$.

*3.3. Meteorological Conditions and Atmospheric Pollutant Concentrations*
3.3.1. Houston Sites

The annual mean for the meteorological variables did not show much variation across the sites (Table S5). All correlations between the pollutants and meteorological variables were highly significant as $p < 0.001$ (Table S6). There were negative correlations seen between the pollutants and humidity, but $PM_{2.5}$ had positive correlations with temperature as opposed to $NO_2$ which had negative correlations (Table S6). The pollutant concentration across the sites was usually highest where the wind speed was at its lowest, which was confirmed by negative correlations seen between the pollution concentrations with the wind speed (Figure A6) (Table S6). The $PM_{2.5}$ was highest at high wind speed for Baytown, but it had high concentrations with high wind speeds from the southeast at Aldine (Figure A6). The $PM_{2.5}$ in both sites was highest with winds from the southeast, but the $NO_2$ in Aldine and Lynchburg was highest with winds coming from the northwest (Figure A6).

3.3.2. London Sites

The annual mean for the meteorological variables showed a little variation across the sites (Table S7). The pollutants exhibited highly significant weak positive or negative correlations with the metrological variables as $p < 0.001$ (Table S7). $NO_2$ and $SO_2$ had a positive correlation with temperature at London Marylebone, but in Thurrock the correlation was negative; the reverse was seen with $NO_2$ and $SO_2$ with humidity (Table S8). $PM_{10}$ in both

sites had negative relationship with temperature, it was positive with humidity in Thurrock and negative at London Marylebone Road. $PM_{2.5}$ had positive correlations with humidity and temperature at London Marylebone Road. The pollutants were highest where the wind speed was low, except $SO_2$ and $NO_2$ at London Marylebone Road (Figure A7). This was confirmed by negative correlations seen between the pollutants and wind speed across the sites, except $SO_2$ (Table S8). $NO_2$ and $SO_2$ at both sites were highest with winds from the southwest (Figure A7). $PM_{10}$ was highest with winds with low to moderate speeds from the east in both sites; this was also seen with $PM_{2.5}$ in London Marylebone.

### 3.3.3. Southampton Sites

The annual mean for the meteorological variables showed a little variation at both sites (Table S7). The pollutant concentrations exhibited highly significant weak positive or negative correlations with the metrological variables as $p < 0.001$, except for $PM_{10}$ with humidity and temperature at Southampton Centre and Southampton A33, respectively (Table S8). The pollutants across the sites had negative correlations with temperature. $NO_2$ and $PM_{10}$ had weak positive correlations with humidity in Southampton Centre, but it was negative at Southampton A33. The $SO_2$ and $PM_{2.5}$ in Southampton Centre had negative correlations with humidity. The $NO_2$ concentration for both sites tended to be higher when the wind speed was low, confirmed by negative correlations between $NO_2$ and wind speed (Figure A7) (Table S8). The highest $NO_2$ concentrations at Southampton Centre occurred at all wind directions, whilst in Southampton A33 it was seen with south-easterly winds (Figure A7). $PM_{10}$ at both sites was highest with easterly winds but with high wind speeds at Southampton A33. $PM_{10}$ had weak negative correlations with the wind speed across the sites. $SO_2$ and $PM_{2.5}$ in Southampton Centre were highest with southerly winds at high wind speeds and easterly winds at low wind speeds, respectively, although they displayed negative correlations with wind speed.

## 4. Discussion

### 4.1. The Drivers of Atmospheric Pollution in Houston

The pollutant concentrations across the selected sites declined over the study period, and the $PM_{2.5}$ concentrations did not exceed the national limits from the late 2000s onwards. This is consistent with other studies [59,60]. This decline in pollutant concentration may be linked to the implementation of national limits for atmospheric pollutants in the 1970 Clean Air Act [59,61]. Houston's population is likely to have a lower risk of adverse health impacts as $PM_{2.5}$ was within the limits [62]. However, air pollution still poses adverse impacts especially to vulnerable populations in Houston [59]. $NO_2$ may have exceeded the limits across the study, leading to adverse impacts on health.

The pollutants were typically higher in Aldine compared to Baytown and Lynchburg Ferry as wind speed was lower in Aldine. The dispersion of the pollutants is lower with low wind speeds compared to high winds speeds, so the pollutants are likely to accumulate in Aldine compared to Lynchburg Ferry and Baytown [41]. The negative correlations between the pollutant concentration across the sites and wind speed is also attributed to this. The high pollutant concentrations at low wind speeds suggest that emissions are from low-level localised sources, e.g., road traffic [41]. Therefore, the high $NO_2$ concentrations seen in Aldine and Lynchburg were likely driven by road traffic from the nearby Aldine Mail Route and Independence Parkway Roads [63]. The $PM_{2.5}$ concentrations were highest at Aldine and Baytown, accompanied by south-easterly winds at high speeds; this could be driven by transport of PM from Baytown oil refinery, and the Port of Houston in the southeast of the city [64].

$NO_2$ concentrations were higher in winter, which is related to greater conversion of nitrous oxide (NO) to $NO_2$ under shallow, stagnant inversion layers. Temperature inversion layers are persistent during high-pressure systems and low temperatures seen in winter, so there is less vertical mixing in the atmosphere and dispersion of $NO_2$ [65,66]. This can be attributed to $NO_2$'s negative correlation with temperature across the sites. Positive

correlations between PM and temperature are seen in Houston due to faster oxidation of volatile organic compounds producing secondary PM at high temperatures [67]. High $PM_{2.5}$ seen in the summer across the sites is attributed to this, as temperatures are higher in summer. $NO_2$ and $PM_{2.5}$ at the sites were higher during weekdays compared to the weekends due to reduced traffic flows and port activity seen on the weekends [68]. Studies have found diurnal patterns in pollutant concentration; the lack of hourly data meant that these could not be studied in Houston [68].

*4.2. The Drivers of Atmospheric Pollution in London*

The $NO_2$ and $PM_{10}$ concentrations did not exceed the national limits in Thurrock compared to London Marylebone where they have continually exceeded the national objectives 2000–2019 (Figure A3). Thurrock's population probably has a lower risk of developing severe health conditions compared to that in London Marylebone [62]. The pollutant concentrations were higher in London Marylebone Road compared to Thurrock, which is likely driven by Marylebone's Road high traffic volumes [69,70]. Studies have found that traffic flow is significantly linked with air pollution due to increased vehicular emissions. However, most of the traffic counts data were not significantly correlated with pollution concentration in London. This was likely due to small sample size caused by the lack of hourly traffic count data covering 2000 to 2019.

The pollutants had negative correlations with wind speed similar to Houston. The average wind speed for London Marylebone Road was similar to that in Thurrock, so it was probably not a confounding factor in contributing to high pollution at London Marylebone. Marylebone road is an urban street canyon which is a narrow road with tall buildings aligned parallel to it on both sites [71,72]. It has the distinction of being the world's most studied road in terms of air pollution and it is likely the most polluted street in London, although this is difficult to prove conclusively [73]. The skimming flow of air is common in street canyons; it can sweep pollutants from above buildings to the ground level, and it reduces the dispersion of pollutants in the area [71]. The $NO_2$ and $SO_2$ concentrations are highest at both sites with westerly winds (Figure A7). This is driven by congestion related to Heathrow Airport and the busy Dartford crossing that are located west of the sites [74]. $NO_2$ and $SO_2$ concentrations are highest at low wind speeds for Thurrock, but they occurred for all wind speeds at London Marylebone Road, likely driven by its urban street canyon structure [75]. Heathrow Airport and the Dartford crossing are situated to west of London Marylebone Road and Thurrock, respectively. Moderate to high concentrations of $SO_2$ and $NO_2$ are seen with south-easterly winds in Thurrock, this is likely driven by emissions associated with the Port of Tilbury, located in southeast Thurrock. However, PM concentrations at both sites were seen in easterly winds with low to moderate wind speeds (Figure A7). This is typical for many UK sites, linked to transport of long-transport PM from Europe [41,75]. The pollutants in Thurrock were negatively correlated with temperature; this was the opposite at London Marylebone. Positive correlations between pollutants and temperature at London Marylebone were not excepted as pollutant concentrations are lower with increased temperature due to enhanced dilution caused by increased thermal turbulence [76].

$NO_2$ concentrations in London exhibited similar seasonal trends to those in Houston. However, high PM concentration was seen in spring, related to the suspension of road dust and long transport of PM from Europe, which are common during spring [77]. PM concentrations were lowest during the summer in London, which could be related to the loss of semi-volatile PM with increased temperatures [66]. The $SO_2$ concentrations across the sites did not exhibit seasonal variation across the sites, which is not consistent with other studies [77,78], but probably because the values are low in any case. The concentrations were typically higher during peak hour on weekdays, which is related to higher traffic volumes.

*4.3. The Drivers of Atmospheric Pollution in Southampton*

Pollutant concentrations across the sites did not exceed the national objectives, so the population of Southampton had a lower risk of severe health conditions compared to that of London Marylebone Road [62]. Nevertheless, air pollution in the city drives emergency respiratory hospital admissions within the city [41]. The $NO_2$ and PM concentrations were typically higher at Southampton A33 since the station was installed in 2016. This is linked to higher traffic flows seen at A33 Redbridge Road near the Southampton A33 monitoring station [62]. However, significant correlations between traffic counts and pollutant concentration were not seen in Southampton, which is linked to the small sample size (Table S5). Southampton A33 is located in proximity to the Port of Southampton's Western Docks, which is a major contributor to $NO_2$ emissions [41].

The pollutants across the sites had negative correlations with wind speed, similar to London and Houston. Wind speed was not the key driver in variation in pollution concentration across Southampton, similar to London, as both sites had similar wind speeds. The $NO_2$ concentrations were highest with low wind speeds from the southeast at Southampton A33, which is likely driven by emissions from the Port of Southampton and Redbridge A33 Road [41,79]. Improvements to dockside air quality at similar locations have been modelled via the use of alternative shipping fuels or via electrification of equipment, with reductions in pollutants by 70–80% [80]. $NO_2$ concentrations were high with winds in all directions for Southampton Centre, so it is likely driven by range of sources. Moderate $NO_2$ concentrations were seen with westerly winds due to emissions from the Port of Southampton and Redbridge A33 Road [41]. The concentrations of PM were highest with south-easterly or easterly winds in both sites. The $SO_2$ concentrations at Southampton Centre were highest with southerly winds; this may be driven by emissions from ships entering the Eastern Docks located in southern Southampton [79]. The pollutants had negative correlations with temperature in Southampton similar to those in Houston and London. Seasonal trends in $NO_2$ and PM concentrations in Southampton were similar to those in London. The $SO_2$ concentrations did not change much with the seasons; they are much lower than those of other pollutants. Pollutant concentrations increased between 05:00 and 08:00 and 14:00 and 19:00, which is linked to high traffic volumes during the rush hours. The pollutants were higher during the weekdays compared to the weekends; reduced traffic flows and port activity is linked to this.

**5. Conclusions**

Authoritative, trustworthy, continual, automatic hourly air quality monitoring is a relatively recent innovation. The task of reliably identifying long-term trends in air quality can therefore be very challenging, as well as complex. Our paper is the first to address this issue for major port cities. The long-term trends for atmospheric pollution in the cities of Houston, London, and Southampton have been successfully mapped, analysed and critically reviewed. Detailed discussions of the temporal and spatial patterns of atmospheric pollutant concentrations alongside discussions of the impacts of meteorological conditions and traffic counts at each location have shed light on trends over a 20-year period. The pollutant concentrations at Houston, Southampton and Thurrock have slowly reduced over time and did not exceed national limits, in contrast to $NO_2$ and $PM_{10}$ concentrations at London Marylebone Road. Contemporaneous studies show that the population in London Marylebone has a high risk of chronic health conditions associated with high pollution. Air pollution has also affected human health in Houston and Southampton. The atmospheric pollutant concentrations were driven largely by local conditions—meteorological, geographical and temporal variation, and traffic flow, namely road and marine traffic at the ports. Sites with the highest traffic flows typically had higher concentrations in London and Southampton, but the presence of the port influenced traffic in Southampton. The emissions from road traffic and the port varied with time, with highest concentrations seen during peak hour periods during the weekends. The metrological conditions influenced

the pollution concentration between the sites and across the sites; pollution concentrations vary across the seasons.

**Supplementary Materials:** The following supporting information can be downloaded at: https://www.mdpi.com/article/10.3390/atmos14071135/s1, Figure S1. United States Environmental Protection Agency (EPA) air quality monitoring stations in: (a) Houston Aldine (b) Houston Baytown and (c) Lynchburg Ferry. Figure S2. Automatic Urban and Monitoring Network (AURN) air quality monitoring stations at (a) London Marylebone road and (b) Thurrock. Figure S3. Map of Automatic Urban and Monitoring Network (AURN) air quality monitoring stations at (a) Southampton Centre and (b) Southampton A33. Table S1. Descriptive statistics of daily maximum $NO_2$ and daily mean $PM_{10}$ concentration for each year from air quality monitoring stations at Houston sites. Table S2. Descriptive statistics of the hourly atmospheric pollutant concentrations for each year from air quality monitoring stations at London Marylebone Road and Thurrock. Table S3. Descriptive statistics of the hourly atmospheric pollutant concentrations for each year from air quality monitoring stations at Southampton Centre and Southampton A33. Table S4. Spearman's Rank Correlation analysis for hourly mean pollutant concentration vs hourly traffic counts between 7:00 hrs and 18:00 hrs on a weekday in March, April, May, July, September and, October over several different years in Southampton Centre, Southampton A33, London Marylebone Road and Thurrock, n = 12 for traffic counts and n = 12 for hourly pollutant concentration. * Signifies a significant correlation ($p < 0.05$). Table S5. Descriptive statistics of daily wind speed and direction, temperature, and humidity at Houston sites from 1 July 2008 to 31 December 2019, n = 365, except for 2008 n = 184. Table S6. Spearman Correlation for daily max $NO_2$ concentration in Houston Aldine and Lynchburg Ferry vs. meteorological variables, and for daily mean $PM_{2.5}$ concentration in Houston Aldine and Houston Baytown from 1 July 2008 to 31 December 2019. For Houston Aldine n = 4036, for Baytown n = 4197, for Lynchburg Ferry n = 4197. Meteorological variables are wind speed and direction, temperature, and humidity. Table S7. Descriptive statistics of hourly wind speed and direction, temperature and humidity at London and Southampton sites from 00:00 hrs 1 July 2008 to 23:00 hrs 31 December 2019, n = 8760, except for 2008 n = 4416. Table S8. Spearman Correlation for hourly mean pollutants concentrations vs meteorological variables at London Marylebone Road, Thurrock, Southampton Centre, and Southampton A33 from 00:00 hrs 1 July 2008 to 23:00 hrs 31 December 2019, n = 100,824. * signifies not a significant relationship ($p > 0.05$). Note: for Southampton A33 pollutant concentrations and meteorological variables are from 00:00 hrs 1 January 2016 to 31 December 2019, with n = 35,064. Table S9. Summary of total hourly traffic counts between 7:00 hrs and 18:00 hrs on a weekday in March, April, May, July, September and, October over several different years in Southampton Centre, Southampton A33, London Marylebone Road and Thurrock (DFT, 2021).

**Author Contributions:** Conceptualization, S.O.-M. and I.D.W.; methodology, S.O.-M., I.D.W. and P.E.O.; validation, S.O.-M., I.D.W., M.D.H., L.M.Z.-R. and P.E.O.; formal analysis, S.O.-M., I.D.W., M.D.H., L.M.Z.-R. and P.E.O.; investigation, S.O.-M.; resources, I.D.W.; data curation, S.O.-M.; writing—original draft preparation, S.O.-M.; writing—review and editing, S.O.-M., I.D.W., M.D.H., L.M.Z.-R., T.J.R. and P.E.O.; visualization, S.O.-M., I.D.W., M.D.H. and P.E.O.; supervision, I.D.W.; project administration, I.D.W.; funding acquisition, I.D.W. All authors have read and agreed to the published version of the manuscript.

**Funding:** This paper has been produced as part of the European Union project "EMERGE: Evaluation, Control and Mitigation of the Environmental Impacts of Shipping Emissions" (referred to as "EMERGE"). The EMERGE project has received funding from the European Union's Horizon 2020 Programme Research and Innovation action under grant agreement No. 874990.

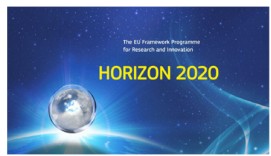

**Institutional Review Board Statement:** Not applicable.

**Informed Consent Statement:** Not applicable.

**Data Availability Statement:** The data presented in this study are available on request from the corresponding author.

**Conflicts of Interest:** The authors declare no conflict of interest. The funders had no role in the design of the study; in the collection, analyses, or interpretation of data; in the writing of the manuscript; or in the decision to publish the results.

## Appendix A

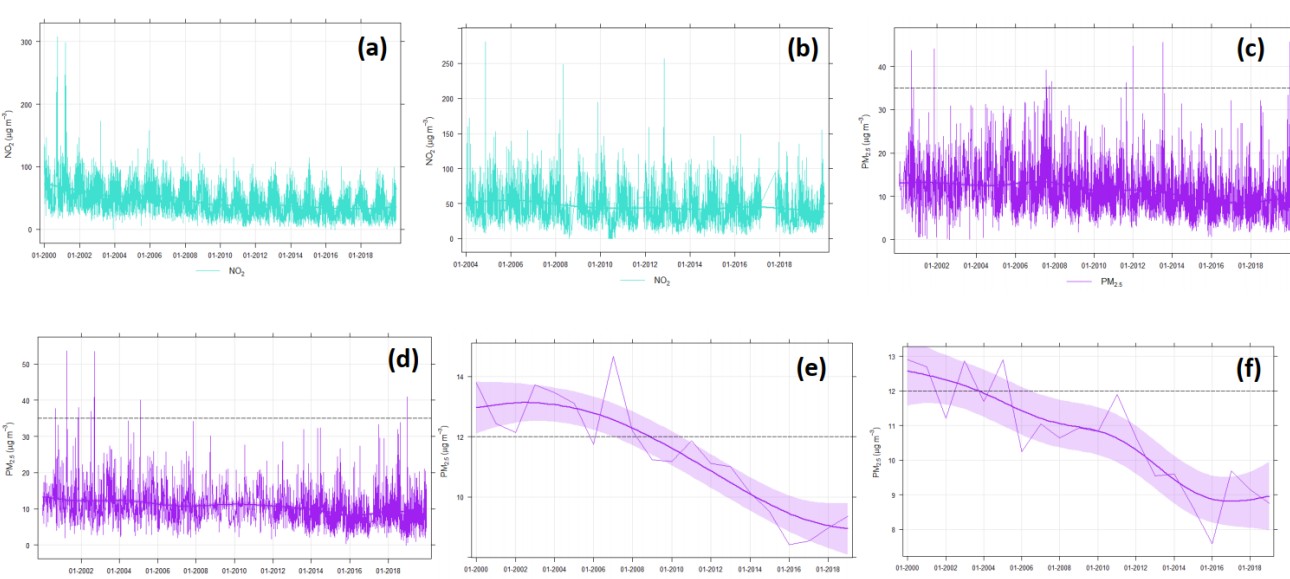

**Figure A1.** Time series plots of daily maximum $NO_2$ concentration and daily mean $PM_{2.5}$ concentration in (**a**,**c**) Houston Aldine, (**b**) Lynchburg Ferry, and (**d**) Houston Baytown from 2000–2019, except Lynchburg Ferry (2004–2019), with 95% confidence interval smooth trendline. The dotted line represents the USA national ambient air quality standard limits for 24 h $PM_{2.5}$ concentration (35 $\mu$g m$^{-3}$). Time plots of annual mean $PM_{2.5}$ concentration for (**e**) Houston Aldine and (**f**) Houston Baytown, with a 95% confidence interval smooth trend line. The dotted line represents the USA national ambient air quality standard limits for 1-year primary $PM_{2.5}$ concentration (12 $\mu$g m$^{-3}$).

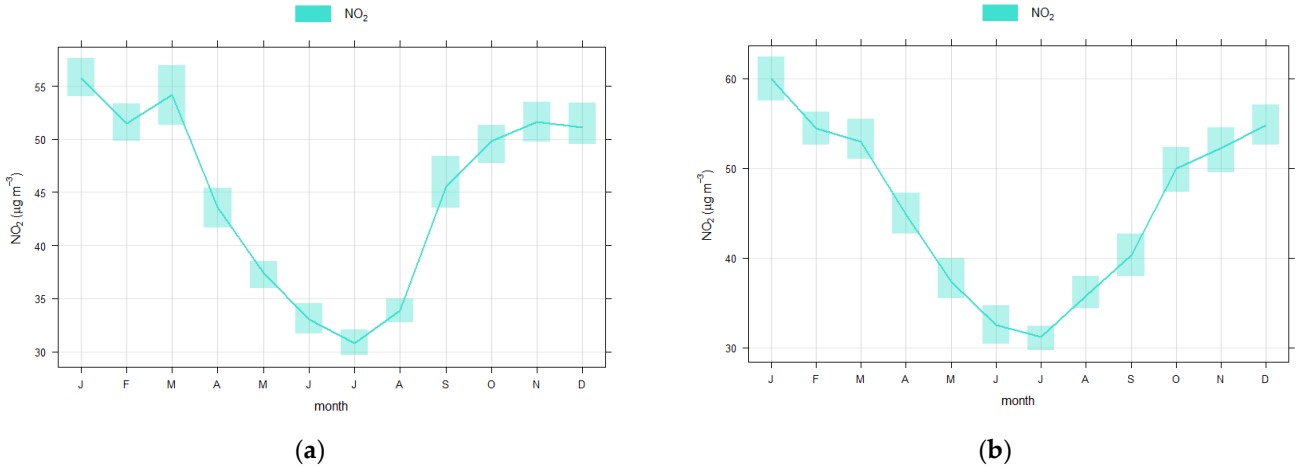

**Figure A2.** *Cont.*

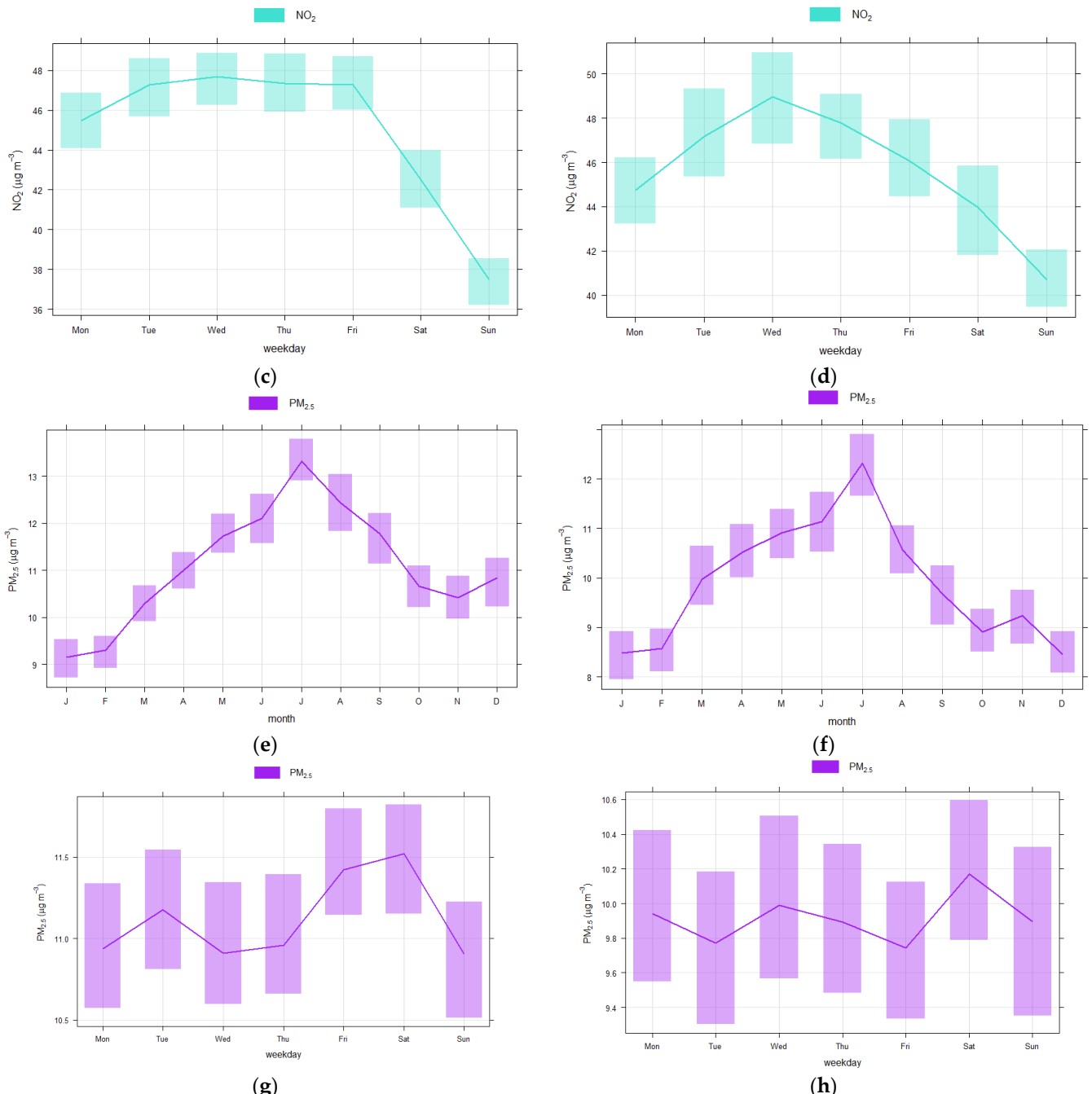

**Figure A2.** Monthly and weekly variation in NO$_2$ and PM$_{2.5}$ concentrations at (**a**) Houston Aldine, (**b**) Houston Lynchburg Ferry, (**c**) Houston Aldine, (**d**) Houston Lynchburg Ferry, (**e**) Houston Aldine, (**f**) Houston Baytown, (**g**) Houston Aldine and (**h**) Houston Baytown from 1 January 2000 to 31 December 2019, with the exception of Houston Lynchburg Ferry (from 2004–2019). Based on daily maximum NO$_2$ and daily mean PM$_{2.5}$ concentration, smooth lines represent the mean and boxes are the 95% confidence interval.

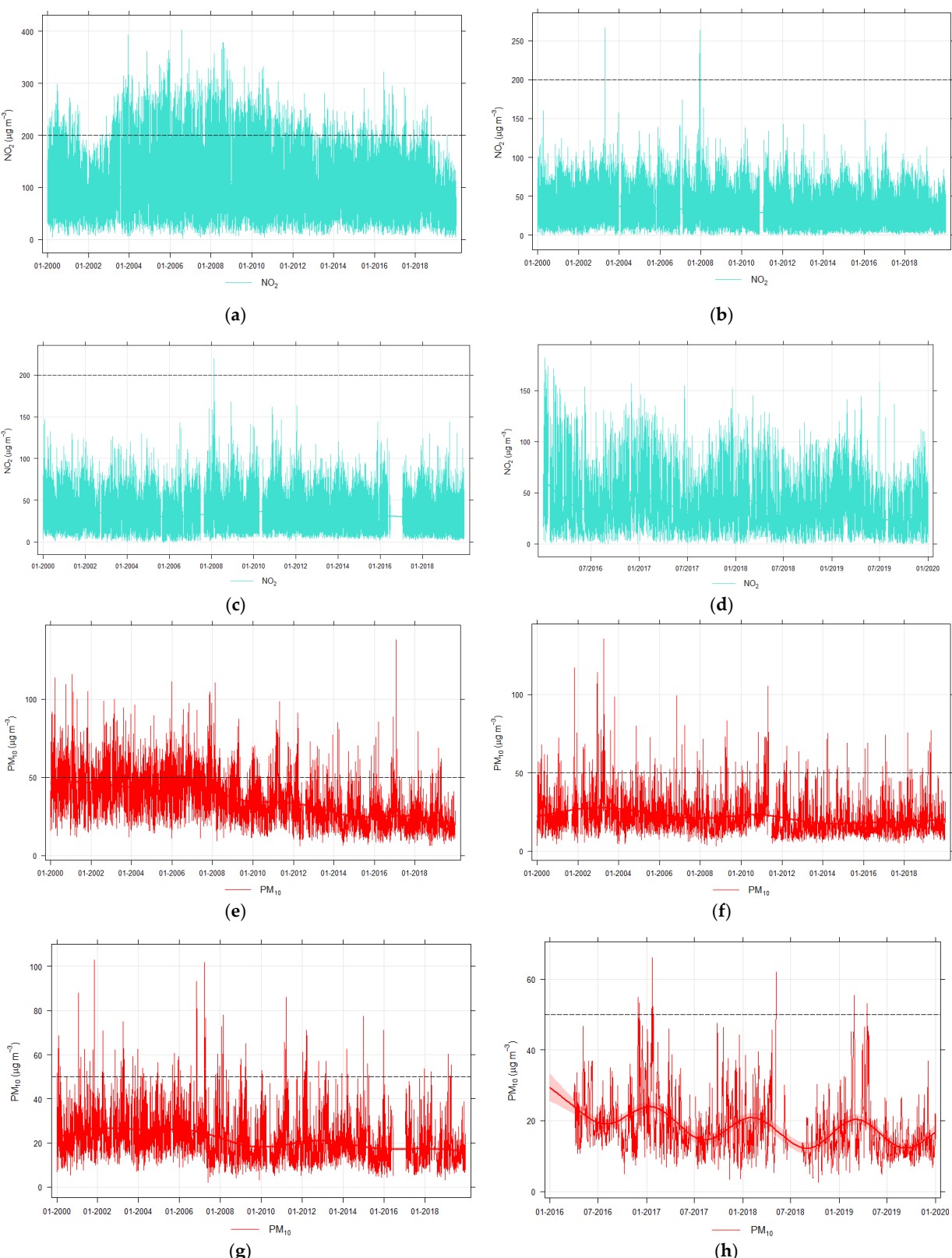

**Figure A3.** Time series plots of hourly mean NO$_2$ and daily mean PM$_{10}$ concentrations at (**a**,**e**) London Marylebone Road, (**b**,**f**) Thurrock, Southampton Centre (**c**,**g**) and Southampton A33 (**d**,**h**) air monitoring sites from 1 January 2000 to 31 December 2019, with a 95% confidence interval smooth trendline. The dotted line represents the UK air quality objectives for 1 h NO$_2$ concentration (200 µg m$^{-3}$) and 24 h PM$_{10}$ concentration (50 µg m$^{-3}$).

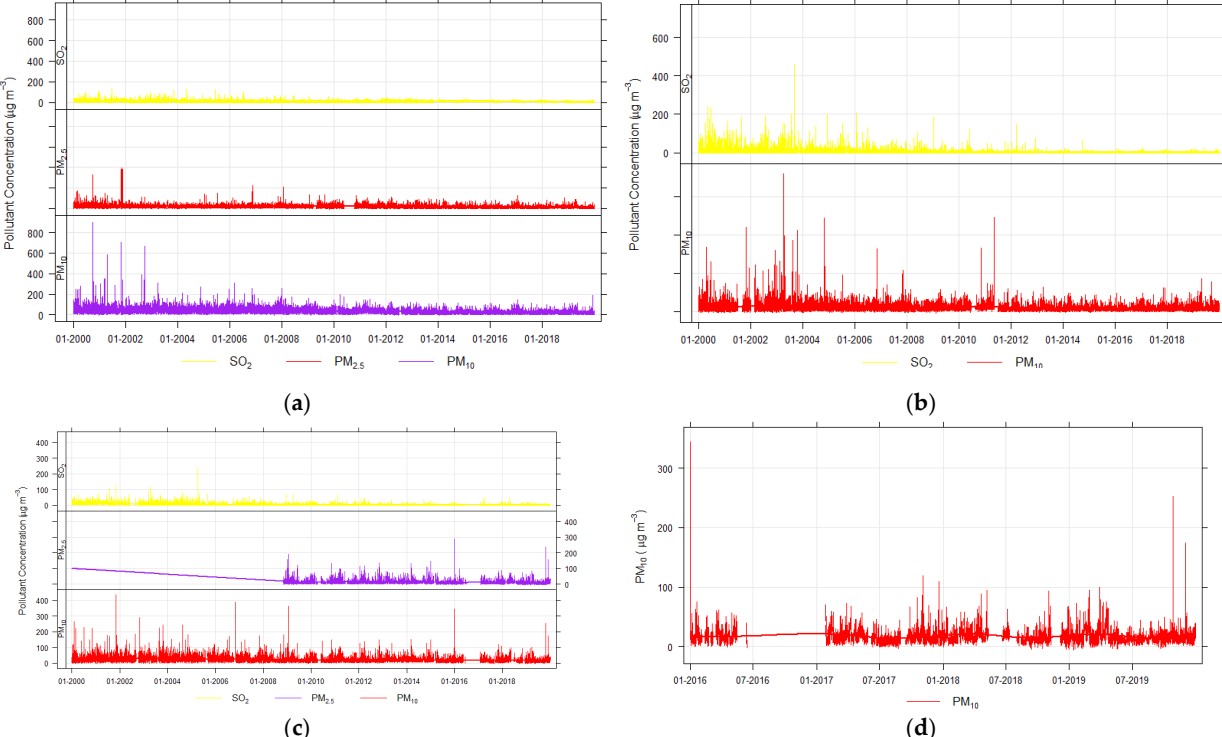

**Figure A4.** Time series plots of hourly concentrations of atmospheric pollutants at (**a**) London Marylebone Road, (**b**) Thurrock, (**c**) Southampton Centre and (**d**) Southampton A33 air quality monitoring sites from 00:00 h 1 January 2000 to 23:00 h 31 December 2019, with a 95% confidence interval smooth trendline.

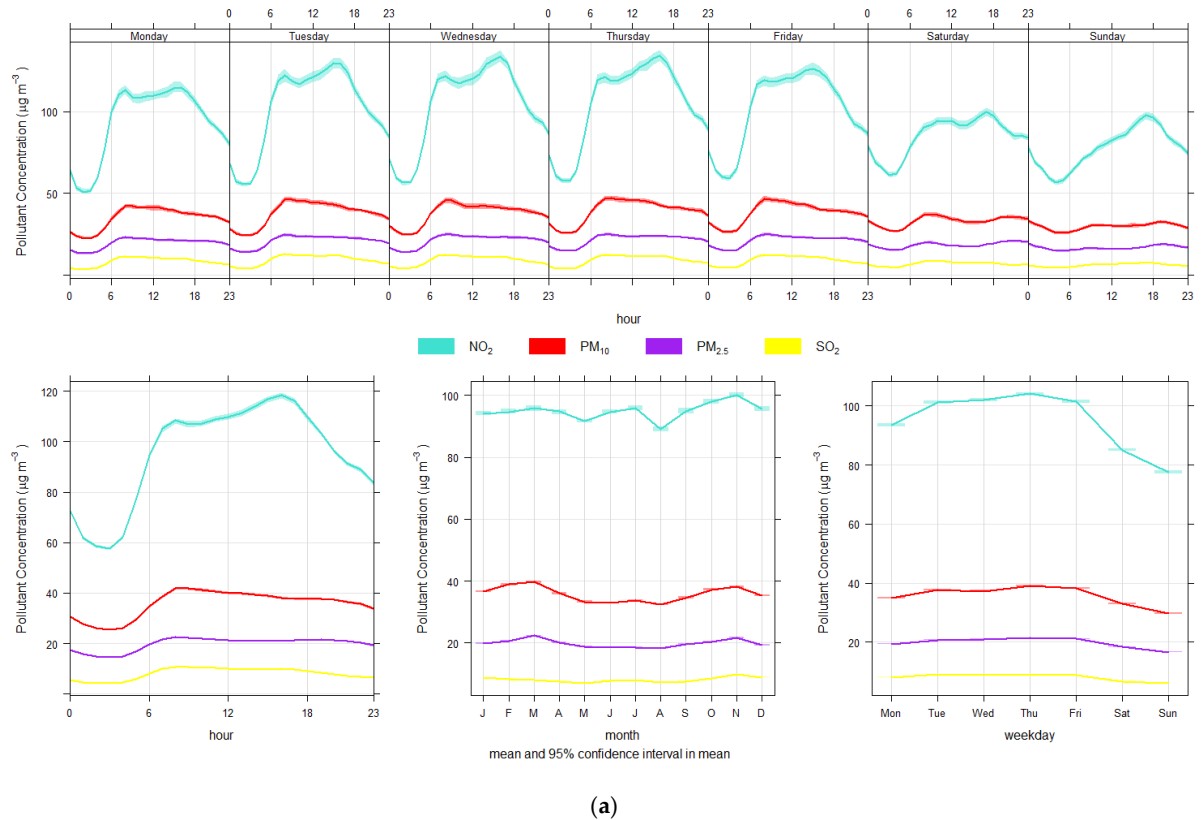

(**a**)

**Figure A5.** *Cont.*

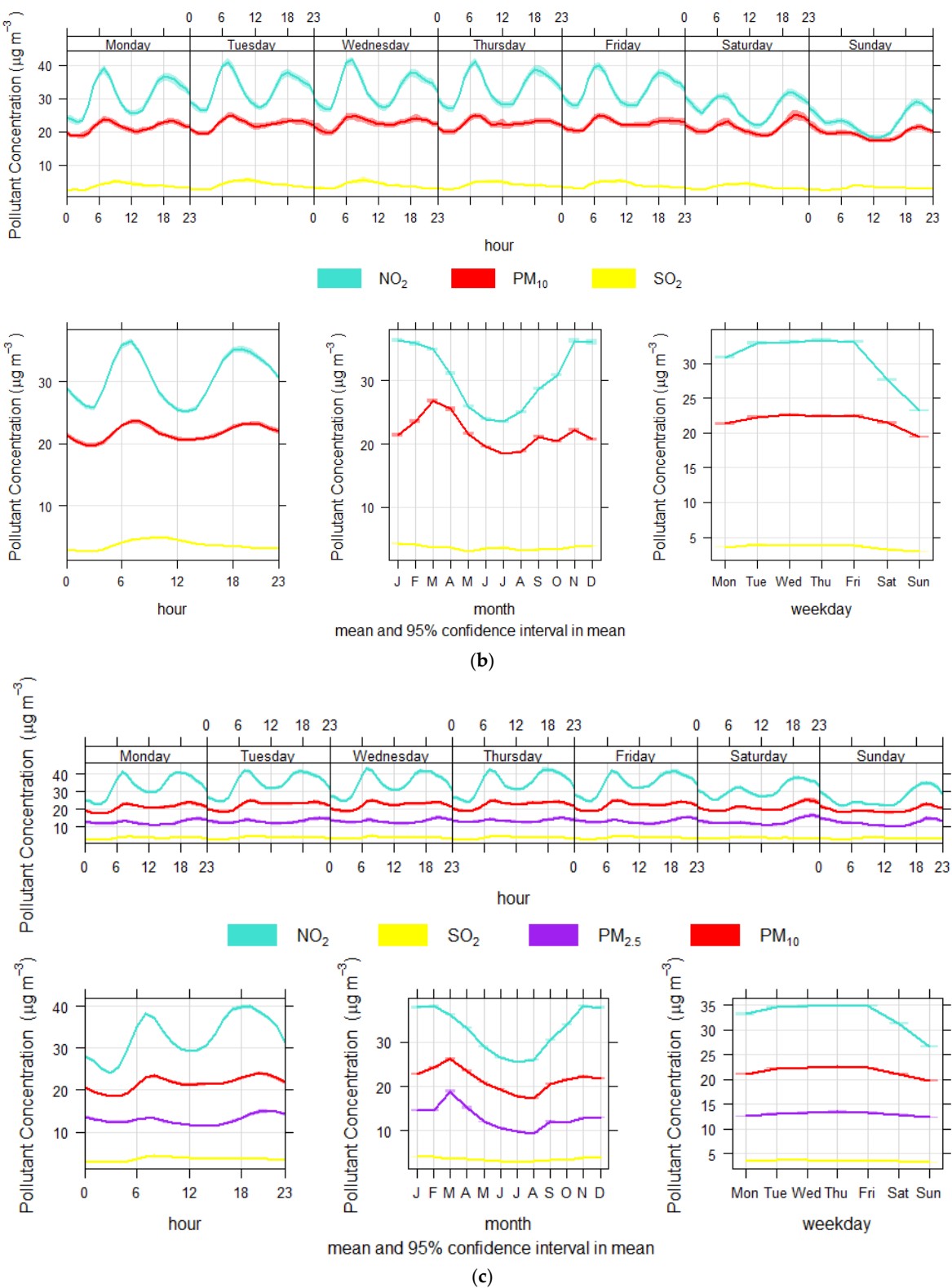

**Figure A5.** *Cont.*

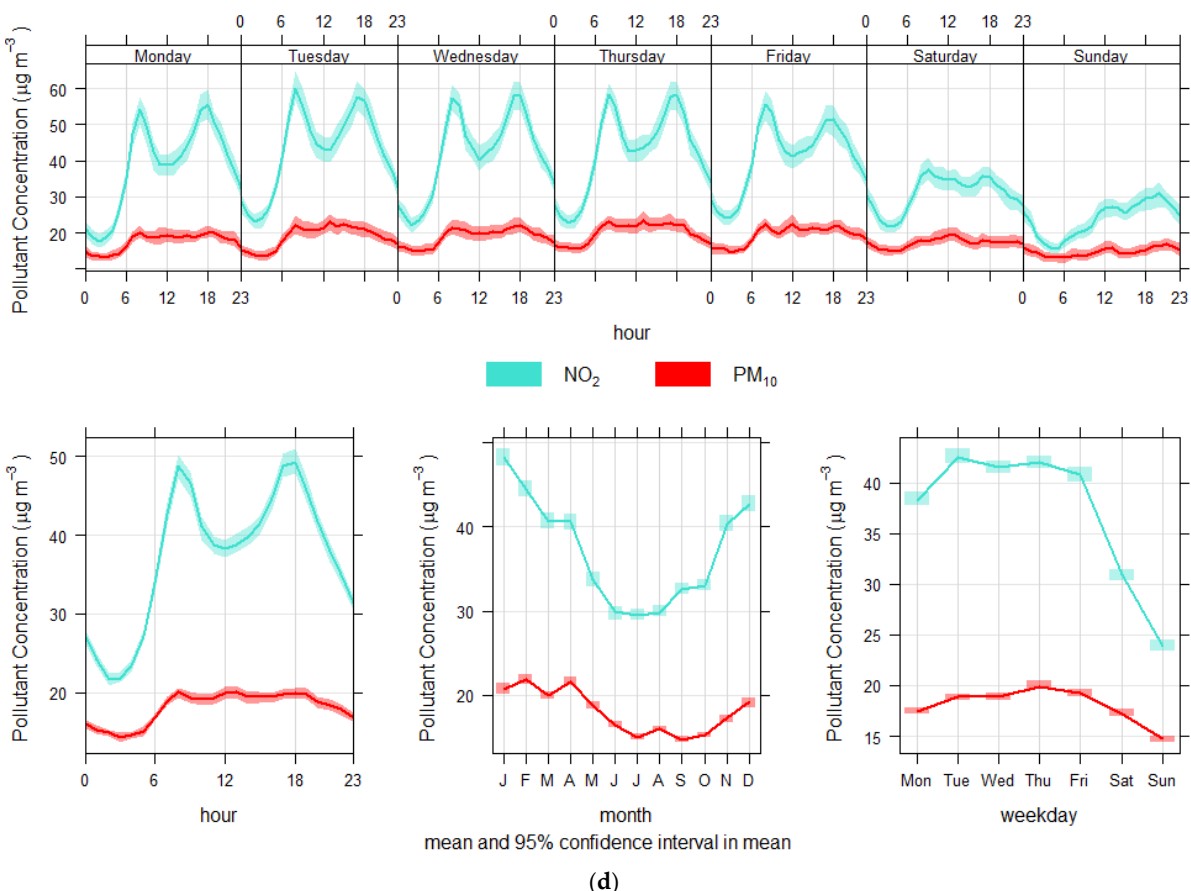

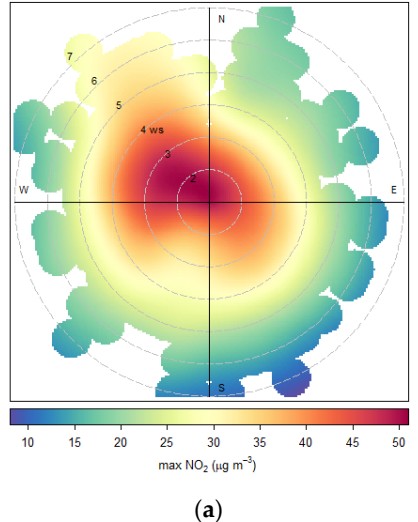

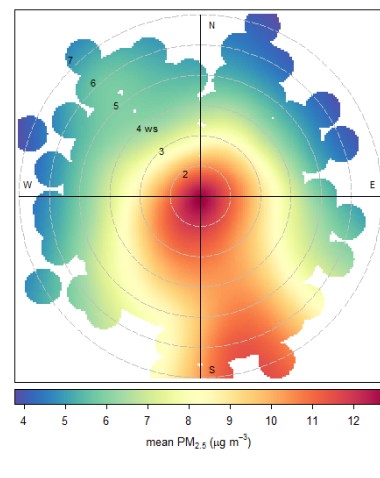

**Figure A5.** Time variation plots of mean pollutant concentrations by hour of weekday, by hour, by month and weekend at (**a**) London Marylebone Road, (**b**) Thurrock, (**c**) Southampton Centre and (**d**) Southampton A33 from 00:00 h 1 January 2000 to 23:00 h 31 December 2019, with smooth line representing the mean and boxes representing the 95% confidence interval.

**Figure A6.** *Cont.*

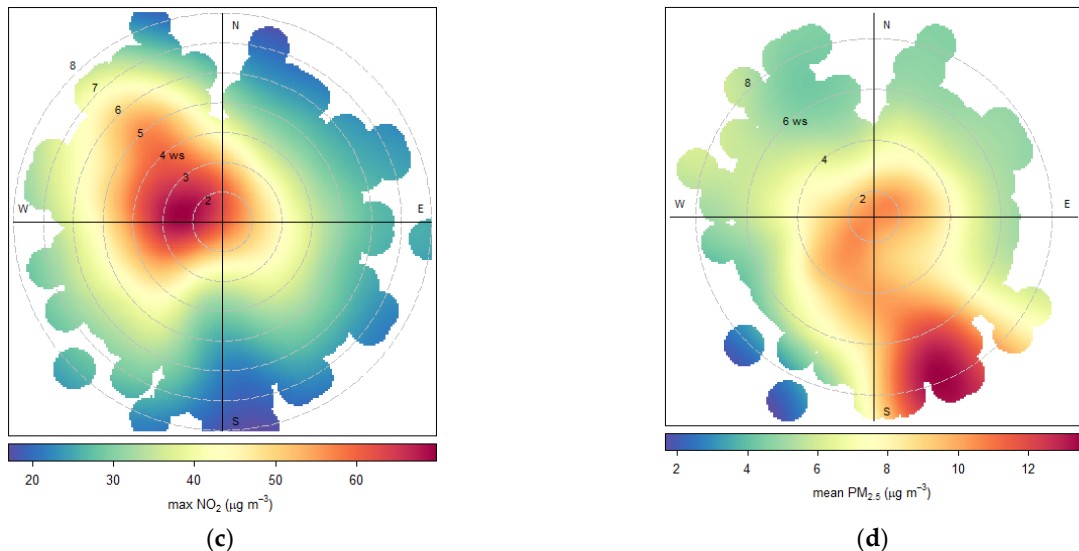

(**c**)  (**d**)

**Figure A6.** Polar plots of daily max NO$_2$ concentration and daily mean PM$_{2.5}$ concentration by wind speed (m s$^{-1}$) and direction in (**a,c**) Houston Aldine, (**b**) Lynchburg Ferry, and (**d**) Houston Baytown between 1 January 2000 and 31 December 2019, except Lynchburg Ferry (1 January 2004 to 31 December 2019). Ws is the wind speed.

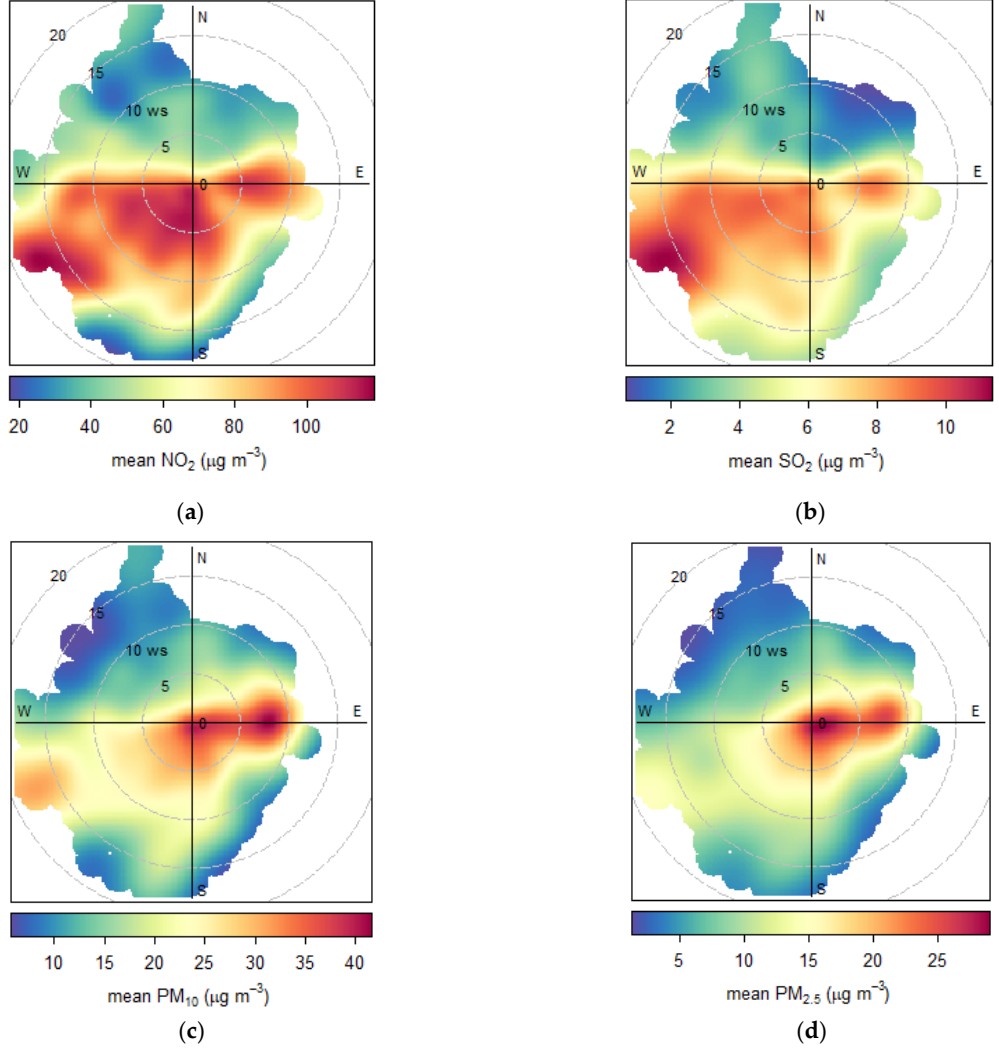

(**a**)  (**b**)

(**c**)  (**d**)

**Figure A7.** *Cont.*

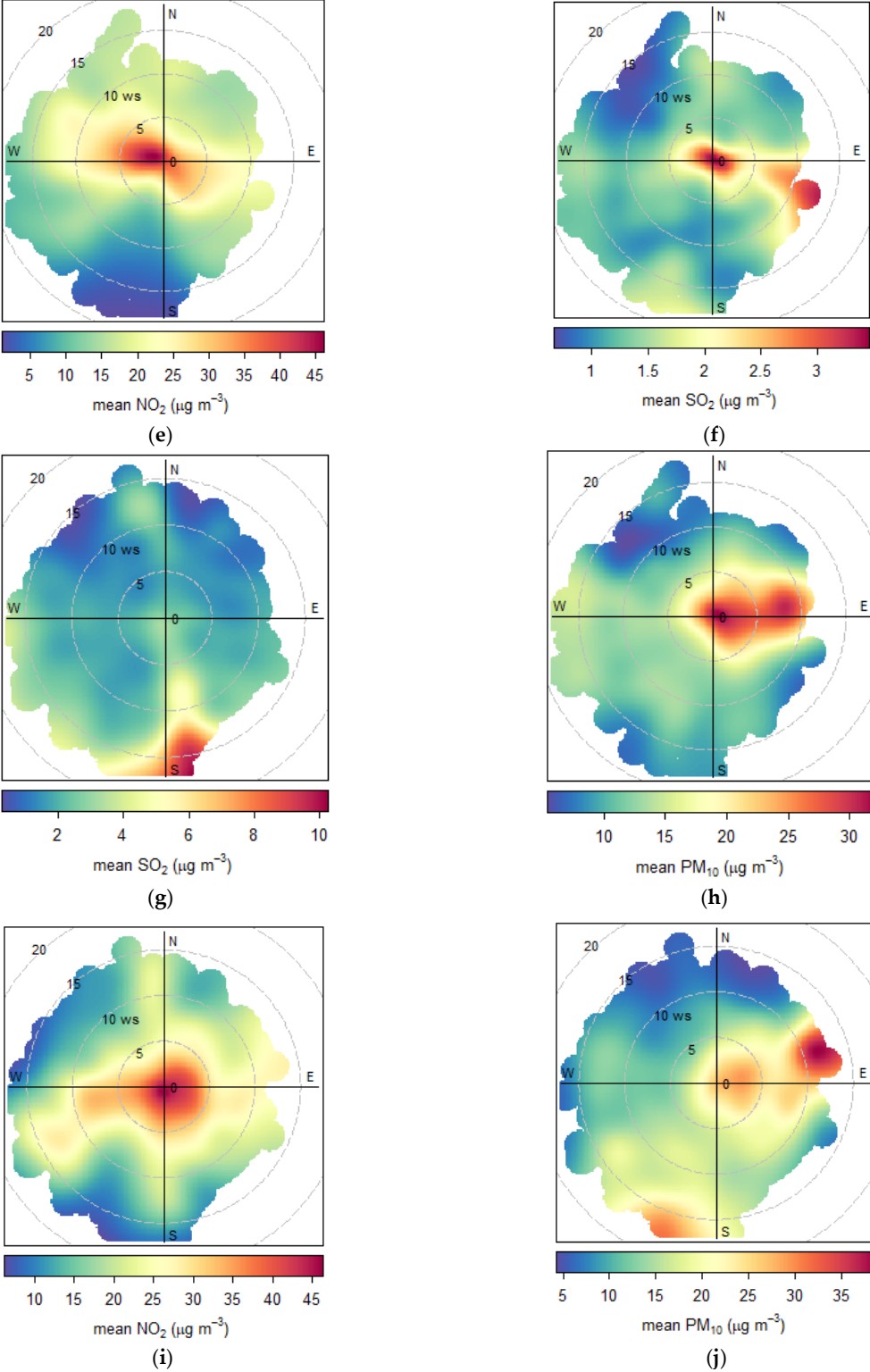

**Figure A7.** *Cont.*

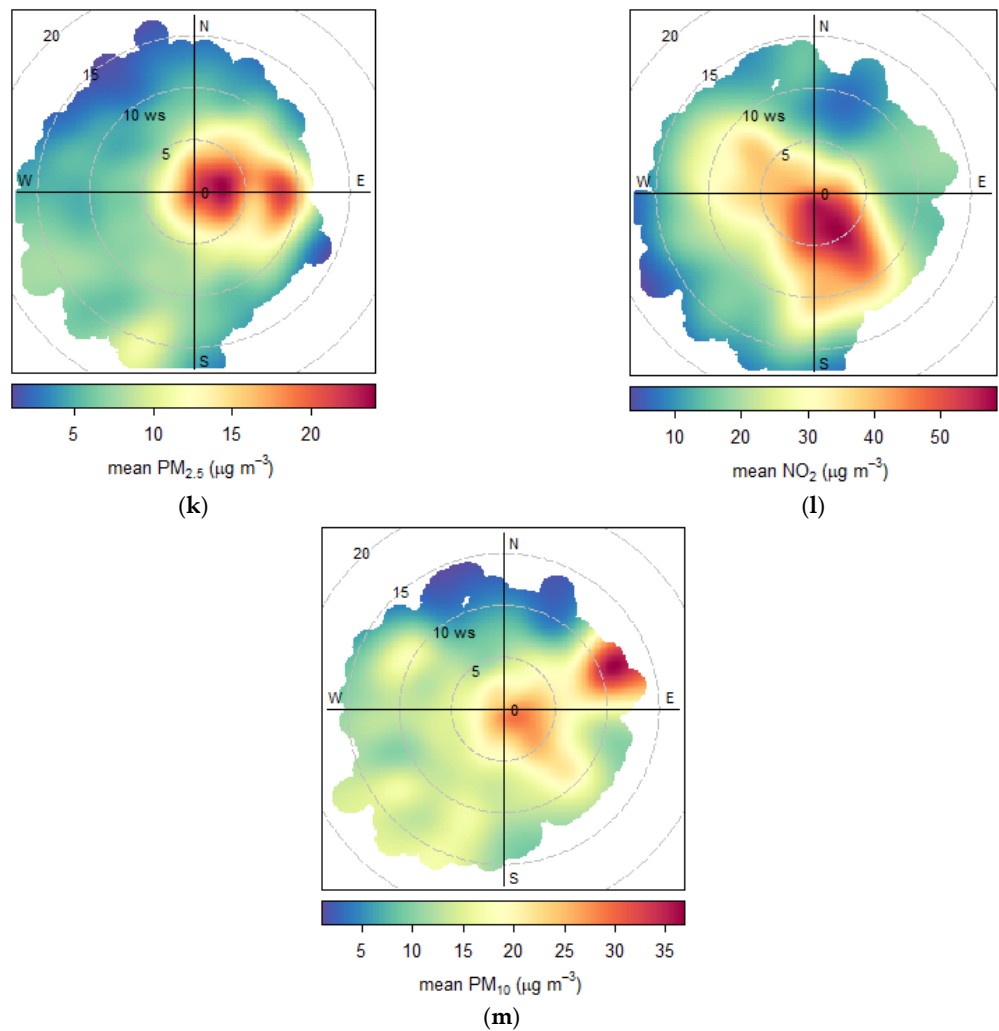

**Figure A7.** Polar plots of hourly mean concentrations of each pollutant by wind speed (m s$^{-1}$) and wind direction at (**a–d**) London Marylebone Road, (**e–g**) Thurrock, Southampton Centre (**h–k**) and Southampton A33 (**l,m**) between 00:00 h 1 January 2000 and 23:00 h 31 December 2019 (except Southampton A33 from 1 January 2016 to 31 December 2019). Ws is the wind speed.

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
