# Peer review of "Atmospheric Pollution in Port Cities"

_atmosphere, doi:10.3390/atmos14071135_

Round 1

Reviewer 1 Report

The paper investigated the long-term trends and drivers of atmospheric pollution in the port-cities of Houston, London, and Southampton 2000-2019

The study is quite interesting.

Some indications are to improve the quality of work are given below:

- more exhaustive abstract on the description of the methodology used in the study and with quantitative indications of the results

- in the introduction the scientific literature and the review should be expanded. These two studies may be helpful:

Campisi et al., 2022 - Locally integrated partnership as a tool to implement a Smart Port Management Strategy: The case of the port of Ravenna (Italy)

Marinello et al., 2021 - Sustainability of logistics infrastructures: operational and technological alternatives to reduce the impact on air quality

- Data processing and analysis section contains a lot of information that is not useful for study (e.g. equation 1). Improve this section

Reviewer 2 Report

The accurate quantification of pollution sources from port cities is important and estimating long-term trends help determine the efficacy of mitigation policies employed. The manuscript is an attempt to quantify and address the above, however, I have some serious concerns regarding its presentation. The authors have used a two-decade-long dataset and extensive data analysis is done but a clear emphasis on findings is warranted here, especially in the discussion section. There are some issues in the figures as well which I have elaborated on below. I think the study is a good attempt in the right direction but needs some improvements prior to its consideration for publication. My detailed comments are listed below:

Major Comments:

1.     It seems that plots were a mere compilation of individual plots which are arranged poorly, is difficult to follow, and lack consistent font sizes (for axis labels, titles, and other information). To be specific, I have the following comments on the figures.

a.     Figs. A1, A3, A4: A general comment is to increase font size of axes labels and titles so that figures are easier to read. A good example is Fig. A5 (b-d).

b.     Fig. A2: box-whisker plot could have been a better representation in my view.

c.     Fig. A3: daily mean lines are barely visible for NO2 plots.

d.     Fig. A4(c) is inverted!

e.     Fig. A5(a-d) has several plots for each alphabetical numbering! These should be properly labeled and captioned.

f.      Figs. A6-A7: The figure is a multi-panel plot (polar plots of pollutant concentration over several cities). The arrangement is very poor, and it seems several individual plots were just pasted to document. These are difficult to follow, as an example authors could have created 4 plots per row with each representing pollutant concentrations, and added rows to represent different sites. The above should be followed with proper numbering and captions.

2.     Almost the entire discussion section is based on existing research articles which is evident with an excessively long reference list for a research article of this size. It is hard to differentiate between findings from this study and what is previously known, which begs an important question on the need/novelty of this article. Authors should use their results to draw conclusions.

Minor Comments:

1.     Why is Houston chosen from the US and not Louisiana or any other port cities for this comparison?

2.     Any reasons for not using mean ± standard deviation in plots?

3.     In the results section (Lines 204-390) several times supplementary materials are cited in paragraphs. As an example, in the first paragraph (section 3.1) authors referred to supplementary materials 6 times and this continues in subsequent sections, instead, authors could have mentioned once at the end of the paragraph “for detailed results refer Table-S1” (assuming each paragraph cites a particular table).

4.     L31-32, 41-42: Citation 5 and 8, either elaborate these sentences or remove them.

5.     L52: “contribute to most of the total emissions in a port city”, this is only true for SO2

6.      L174: “NO2 data was converted to μg m-3 to maintain consistency with the rest of the data”, typically gaseous pollutants are measured in units of ppb/ppm/ppt, I am not sure what is the need to maintain this consistency. I understand one utility is for plotting (all pollutants. Concentrations are in the same units). However, using a linear axis would hide some of the details present in the diurnal/seasonal cycle of some of the pollutants, e.g., in Fig. A5 (a-c) due to the selected linear axis range (y-axis) it is hard to see diurnal/seasonal changes in SO2.

7.     Fig. A5 (a-d): Are 95% variations in monthly and weekday plots this low (refer to ‘vertical thickness’ along the y-axis)? Is this due to a lack of data? What do you mean by variation along the x-axis? The Mon-Sun plots show a lot of variations in mean pollutant concentrations but in diurnal plots there is too little variation, why?

8.     L207-208: “The trends in the atmospheric pollutant concentration in Houston cannot easily be compared with the UK sites, as it was averaged over a 24-hour period compared with a 1-hour period for UK sites.”, the data for UK sites could be converted to the 24-hour average for comparison?

9.     L399-402 is not clear, what do authors imply here?

10.  The reference list is just too long for the size of the manuscript, authors may consider removing some of them.

The overall quality of the English language is fine, I have one minor comment which I have indicated as Minor comment 9 in my review report

Reviewer 3 Report

The study is quite interesting, and worthy to publish, after some additions: 

- there are some mistakes like for example line 50 "associated with to increased..." please review that and others that are not difficult to spot with a fresh eye. like i had to read well to understand what is 2000-19, i would suggest just say 2000-2019, makes more straighforward reading...then in 3.1.2 you used 2000-2019, please stay consistent.  line 399 also: 

may have exceed the limits...

- is your study the only one that reports port emissions influence on urban AQ? please add a few sentences on the novelty of your study. and add maybe some literature on emission studies from ports

- in conclusions you say: "The metrological conditions" but i think you mean meteorological conditions...

- Figure 1 - site maps with study sites - could be helpful to understand  the distances between sites? in one case the distance is 37km, isnt it quite far? Found it in the supplementary files.. ok, the question about the distance still stands. plus it is over the  water. Any ideas how to justify that?

- Why the figures in the main file are numbered A1, etc? should just be "Figure 1"

- the time stamp on NO2 data one site is 24-hr the other 1hr... while we have health standards for 1hr and 1 year. Maybe get yearly averages to be able to compare, for comparison sake?

-to better compare the graphs of polltuion concentrations i would put all axes in the same scale range, for two sites but same pollutant, for ex fig 1, etc, the scales changes for diff sites. why? more difficult to compare, no?

in conclusions, how can you claim this: "However,   air pollution has still affected human health in Houston and Southampton". was a health study made here? i dont see it.

Round 2

Reviewer 1 Report

All suggested revisions have been incorporated into the paper, improving its quality

Reviewer 2 Report

The authors have included most of the comments. However, I expect issues related to figures in the manuscript must be taken care of prior to its publication.